# Enhanced Feature Pyramid Vision Transformer for Semantic Segmentation on Thailand Landsat-8 Corpus

Kritchayan Intarat [1], Preesan Rakwatin [2] and Teerapong Panboonyuen [1,3,4,*]

1   Department of Geography, Faculty of Liberal Arts, Thammasat University, 99 Moo 18, Phahonyothin Rd., Khlong Luang, Khlong Nueng, Pathum Thani 12121, Thailand; intaratt@tu.ac.th
2   Digital Economy Promotion Agency, 80 Soi Ladpro 4, Ladprao Rd., Chom Phon, Chatuchak, Bangkok 10900, Thailand; preesan.ra@depa.or.th
3   Department of Computer Engineering, Faculty of Engineering, Chulalongkorn University, Phayathai Rd., Pathumwan, Bangkok 10330, Thailand
4   MARS—Motor AI Recognition Solution, Phayathai Rd., Silom, Bangkok 10500, Thailand
*   Correspondence: teerapong.panboonyuen@gmail.com

**Abstract:** Semantic segmentation on Landsat-8 data is crucial in the integration of diverse data, allowing researchers to achieve more productivity and lower expenses. This research aimed to improve the versatile backbone for dense prediction without convolutions—namely, using the pyramid vision transformer (PRM-VS-TM) to incorporate attention mechanisms across various feature maps. Furthermore, the PRM-VS-TM constructs an end-to-end object detection system without convolutions and uses handcrafted components, such as dense anchors and non-maximum suspension (NMS). The present study was conducted on a private dataset, i.e., the Thailand Landsat-8 challenge. There are three baselines: DeepLab, Swin Transformer (Swin TF), and PRM-VS-TM. Results indicate that the proposed model significantly outperforms all current baselines on the Thailand Landsat-8 corpus, providing F1-scores greater than 80% in almost all categories. Finally, we demonstrate that our model, without utilizing pre-trained settings or any further post-processing, can outperform current state-of-the-art (SOTA) methods for both agriculture and forest classes.

**Keywords:** deep learning; pyramid vision transformer; Landsat-8; satellite image; attention

## 1. Introduction

Recently, the relevance of remote sensing's semantic segmentation has increased. In particular, autonomous semantic segmentation has been investigated in the context of remote sensing research [1–3]. Over the years, a whole spectrum of autonomous driving, automated mapping, and navigation applications have been achieved. For instance, computer technology has transformed deep learning (DL). Besides, numerous procedures exist among current convolutional neural networks (Convnets/CNNs), viz., dual attention deep fusion semantic network [4] and self-attention for semantic segmentation [5]. Due to their potential, they have gained a lot of attention. By applying remote sensing data, exact semantic segmentation can be attained. However, there are concerns regarding accuracy.

In the fields of agriculture and urban segmentation, there are considerable deep neural network models [2,6], such as global convolutional networks with large kernel improvements [7], Deeplab image segmentation [8], mask R-CNN [9], Bilateral segmentation network [10], and Criss-cross attention [11]. These deep architectures, which are made up of layered convolution blocks, have been designed for semantic recognition. Due to processing costs, the use of kernel maps has declined.

It is acknowledged that encoder networks can learn more meaningful visual theories within a steadily increasing receptive area. This situation remains challenging because of a region's limited size in the input that provides the characteristics. Transformers focus their self-attention on these receptive fields, requiring dense, high-resolution predictions.

This effort was motivated by the fact that architecture has not comprehensively employed different feature maps from the convolutional layer or any attention blocks.

According to dense image undertakings of semantic segmentation and detection, several works use ViT models. ViT is a deep learning model that concentrates on image identification and uses an attention mechanism. Due to image classification, some ViT models incorporate a transformer model and non-overlapping image patches. ViT is able to achieve an impressive speed swap on practically all image visualization workloads, corresponding to prior networks. ViT's results on image visualization schemes are promising. Our proposed method, the vision transformer [12] or ViT, regarding these improvements [13–17], is the most relevant. However, ViT is inappropriate for low-resolution kernels and a quadratic increase in image size sophistication. DeiT [18] offers several training strategies to improve and receive efficient outcomes, even when utilizing the ImageNet-1K corpus. Herein, this paper focuses on general-purpose implementations rather than semantic segmentation. ViT [19,20] models appear to offer the best performance–accuracy trade-offs with such strategies as computer vision.

It is noted that prior transformer networks frequently incurred substantial computational expenses. For example, costs amassed by the pyramid vision transformer (PRM-VS-TM) [21] were quite huge. SwinTF, on the other hand, was able to overcome computational problems: expenses were proportional to the size of the image. SwinTF is seen to have increased the model's accuracy by controlling it regionally and boosting receptive fields that are favorably associated with visual inputs. SwinTF is most efficient, exhibiting SOTA performance—e.g., MeanIoU and average precision in COCO object detection and ADE20K image labeling.

In this paper, the adaptable backbone for dense prediction without convolutions known as "PRM-VS-TM" has been improved by enhancing (E) and incorporating attention mechanisms that apply across various feature maps via "E-PRM-VS-TM". Many problems had to be overcome in devising a refined shrinking pyramid and spatial-reduction attention (SRA) method, enabling PVT to adapt and learn high-resolution and multi-scale features. Similarly, an end-to-end object-detected procedure can be constructed without convolutions and handcrafted components, e.g., dense anchors and non-maximum suppression.

The effectiveness of this work is demonstrated by the experimental results on satellite semantic segmentation collections, including the Landsat-8 remotely sensed data (Thailand). Results prove that E-PRM-VS-TF can override prior encoder–decoder networks [22] when employing satellite images associated with transformer models [19,21] after acquiring *Precision*, *Recall*, and *F*1 scores, consecutively.

## 2. Data Collection

There is just one primary data source in our trials: Thailand's Landsat-8 data, covering the area of Nan Province. Nan is located in upper-northern Thailand, bordering Lao's Sainyabuli Province. Nan Province in northern Thailand is a beautiful place with a rich environment, history, and culture. With its tranquil environment and natural charms, the region has grabbed the hearts of both locals and tourists. The picturesque Old Town of Nan City and the magnificent hiking trails and lookouts in Doi Phu Kha National Park are well worth visiting in Nan Province.

The province is located in the secluded Nan River valley, flanked by wooded mountains in the west and the Luang Prabang Range in the east. The tallest mountain is Phu Khe, which stands at 2079 m and is located northeast of Nan, near the Laos border. The total forest area of the province is 7436 sq km (2871 sq mi), accounting for 61.3 percent of the total land area.

Land uses consist of agriculture, forest, urban, water, and miscellaneous; and are represented by yellow, green, red, blue, and brown in Figure 1. In this experiment, the Landsat-8 data comprised 1220 datasets divided into 800 training, 220 validation, and 200 testing sets.

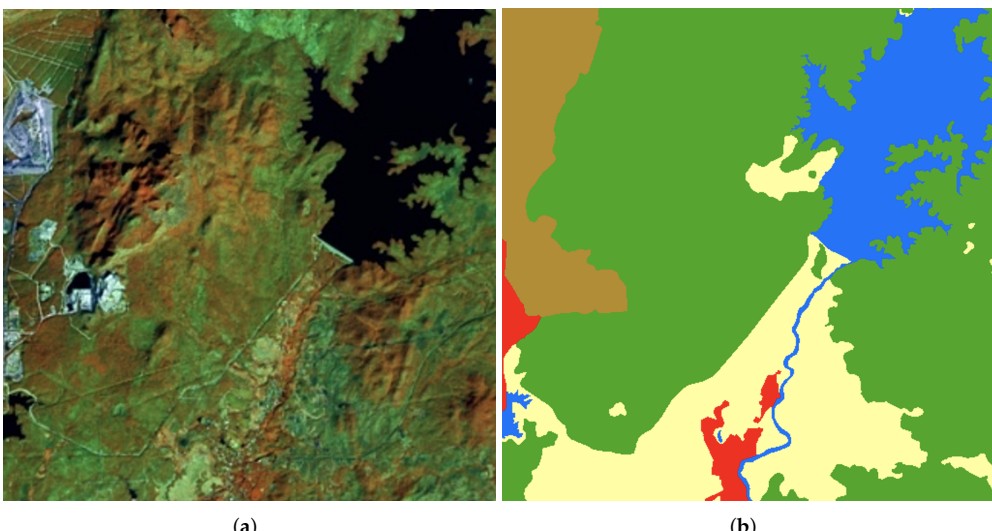

(**a**)                      (**b**)

**Figure 1.** The sample of input image (**a**) and the target label (**b**) from the Landsat-8 corpus.

## 3. Proposed Method

*Pyramid Vision Transformer-Based Semantic Segmentation*

In Figure 2, the enhanced pyramid vision transformer (E-PRM-VS-TM) follows a sequence-to-sequence vector with transformers [21,23] along with a corresponding output vector with input vector fabrication, i.e., the natural language processing (NLP) network.

The previous SwinTF model [19] concentrates on the associations between two tokens or image patches with the rest of the tokens being computed. SwinTF concentrates on the quadratic complexity of image patches' numbers, which results in an improper amount of images, thereby requiring a large number of tokens when reaching the softmax layer.

Transformers have been recognized for their exceptional long-range modeling resiliency. The vision transformer (ViT) [24] has had tremendous success in picture segmentation as the first pure transformer for vision. Such transformers operate "object detection" and offer a learnable query-based approach for instantly retrieving entity data. ViT transformers employ a dual-path transformer having a global memory path and are efficient in semantic segmentation. ViT transformers utilize a hierarchically organized transformer encoder with no positional encoding and a lightweight MLP decoder that deliver excellent performance while reducing computation time. The efficient transformer design can cover many perspectives, such as attention patterns or low-rank approaches.

A transformer network is fit for multi-object segmentation and very-high resolution images. ViT focuses on image recognition, utilizing the mechanism of attention. Accordingly, the transformer network can decide the four crises by referring to ConvNet designs.

Compared to the encoder output, the targeted-feature map is much larger. We ought to perform row/column-wise positional encodings when aligning the query sequence with the row/column flattened key/value sequence. While the aforementioned expresses the mechanism of depicting standard positional encodings with flattening, applying the equivalent concept of row/column-wise expansion is also manageable to expand comparable positional encodings.

Our purpose is to present the pyramid's frontend into the transformer network to forge multi-scale feature maps for dense prediction, e.g., image segmentation. In Figure 2, an overview of "E-PRM-VS-TF" is illustrated. A $1D$ vector embeddings sequence: $z \in R^{L \times C}$ as input, where $L$ represents the vector's length and $C$ denotes the hidden kernel size. Consequently, the image sequence is obliged to adjust the image's input layer, $x \in R^{H \times W \times 3}$, into $Z$.

In Figure 3, the output of attention is described. NLP concerns the interaction between computers and human language in order to process and analyze a large amount of matured language. Accordingly, the SwinTF, as described in Figure 2 allows a $1D$ sequence of

vector embeddings $z \in R^{L \times C}$ as input: $L$ is the length of the vector, and $C$ is the hidden kernel size. The image sequence is consequently obliged to modify an input layer of image $x \in R^{H \times W \times 3}$ into $Z$.

The traditional SwinTF model [19] focuses on the relationship between a token (image patches); the other tokens are calculated. ViT focuses on the quadratic complexity concerning the number of image patches. It is unsuitable for many image problems, requiring an immense set of tokens for the softmax layer.

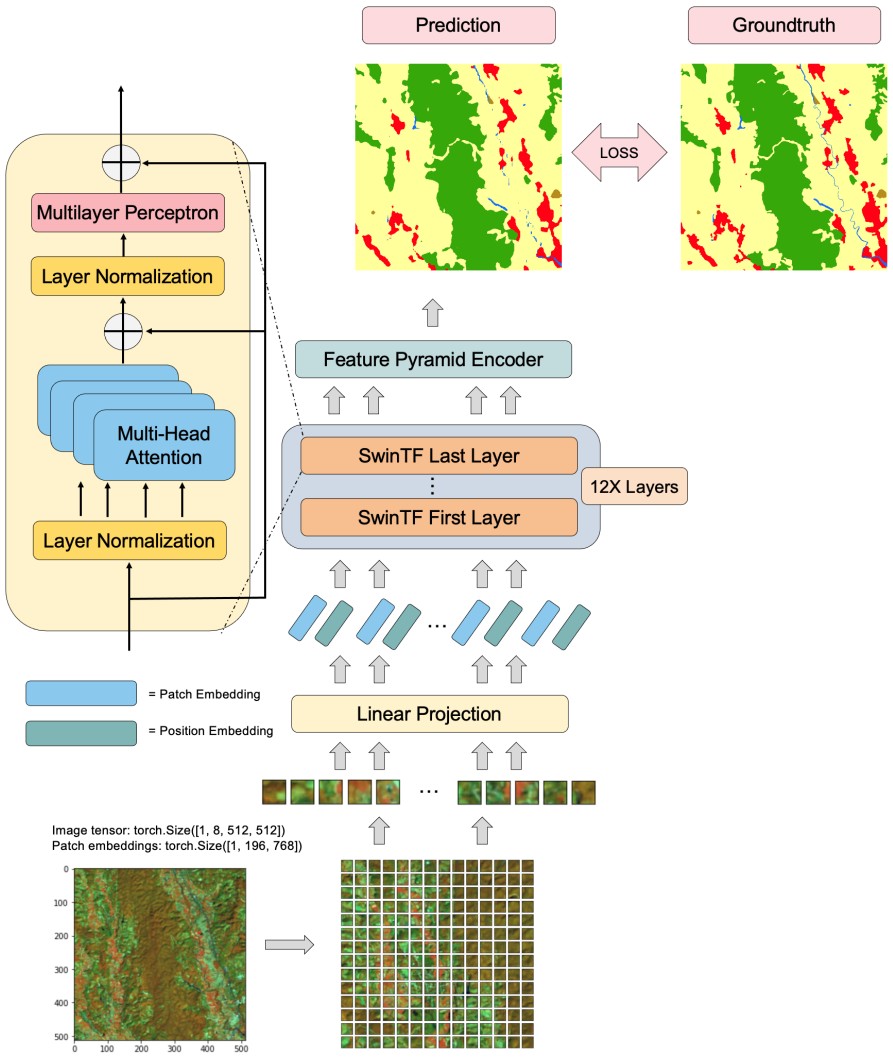

**Figure 2.** The architectural structure of our proposed E-PRM-VS-TF.

To construct the ground truth per mask for semantic segmentation, the segmentation map for each image is decomposed into a set of N ground truth label-mask pairs. This procedure adopts the loss from classification and mask to optimize the probabilities to train our model. According to each layer $I$, the input to self-attention is represented as a triplet of $(query, key, value)$, and computed from the input $Z^{l-1} \in R^{L \times C}$ as in Equation (1) below:

$$query = Z^{l-1}W_Q, key = Z^{l-1}W_K, value = Z^{l-1}W_V \tag{1}$$

where $W_Q/W_K/W_V \in R^{C \times d}$ denote the learnable weights of three linear projection vectors and $d$ represents the dimension of $(query, key, value)$. Self-attention (SA) (Figure 3) is denoted as:

$$SA(Z^{l-1}) = Z^{l-1} + softmax(\frac{Z^{l-1}W_Q(ZW_K)^T}{\sqrt{d}})(Z^{l-1}W_V) \tag{2}$$

MSA clearly calculates a reckoning with m self-supporting SA actions and projects their concatenated outputs: $MSA(Z^{l-1}) = [SA_1(Z_l - 1); SA_2(Z_l - 1); \ldots; SA_m(Z_l - 1)]W_O$. Where $W_O \in R^{md} \times C$. $d$ is typically set to $C/m$. An MLP module transforms the output of MSA with a residual skip as the output layer and expressed it as:

$$Z^l = MSA(Z_{l-1}) + MLP(MSA(Z^{l-1})) \in R^{L \times C}. \tag{3}$$

A global pooling layer from each branch was utilized to acquire globally contextual data in order to place it via a linear transformation ahead of the bilinearly upsampling process to fit the feature dimension. The short path duplicates the input feature and pastes it when all the textual information indicates an output. All developed features are preferably attached to generate the final segmentation map. Dimensionality is reduced using a learned linear transformation and results in a row-wise positional encoding that aligns with row-wise flattening due to a similar code from each row. The related column-wise flattening can use a similar column-wise positional encoding.

Grouping and pooling complement each other. In the group, each query approaches a tiny portion of the flattened sequence at its earliest level. In the pool, each query approaches a small part of the flattened sequence at its actual level. On the other hand, the query uses pooled features in the pooling to access the entire sequence at a coarse level. Output can balance computational costs and representation ability by combining both.

Subsequently, more analogies between our model and other SOTA networks were constructed. In comparison, our proposed method can be employed as an adaptable plug-in decoder for highly-accurate, dense predictions of semantic segmentation.

Lastly, a low-power instrument was applied to create a model inferring declining resource utilization. In other words, the efficient network designs regarding deep learning methods are sufficient to preserve computing resources, promoting deep learning techniques in remote sensing strategies. A normalized layer was operated prior to MSA and MLP modules. $Z^1, Z^2, Z^3, \ldots, Z^{L_e}$ were defined as the transformer vectors' weights.

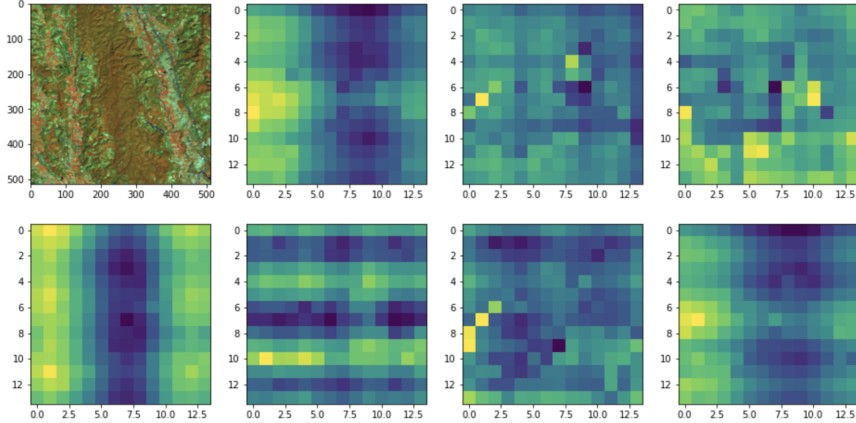

**Figure 3.** The visualization of attention: each cell reveals the cosine similarity between its embedding and other pairwise embeddings.

For deep learning and environmental configurations, the first 70% of training iterations, i.e., 0.25 stochastic depth dropout was employed; for the last 20%, the dropout ratio increased to 0.6. As regards the multi-scale flipping examination, testing scales of 0.5, 0.75, 1.0, 1.25, 1.5, and 1.75 were provided together with random horizontal flips using typical techniques, as described in the literature throughout all the experiments' training, e.g., [12,19,23,25]

Regarding the deep learning (DL) environmental setup, "PyTorch v0.12" was created as an end-to-end open-source platform. All experiments were performed on computer systems with the Intel® Xeon® Scalable 4210R (10 core, 2.4 GHz, 13.75 MB, 100 W), 256 GB

of memory, and the NVIDIA GeForce RTX™ 2080 Ti Integrated with 11GB GDDR6 352-bit memory interface ×2 cards.

A learning rate (LR) schedule with stochastic gradient descent (SGD) was implemented to optimize loss function ensuring sufficient smoothness. For all the trials on the three datasets, sequentially, weight decay and momentum were sealed to 0.15 and 0.85. As for the Thailand Landsat-8 corpus, the initial LR was set to 0.0001. Finally, batch normalization was used in the fusion layers, and batch size of 16 was selected. The images were shrunk to a side length of 496 pixels.

## 4. Experimental Results

According to Table 1, the proposed E-PRM-VS-TF is compared to its counterparts: DeepLabV3 [8], Swin-TF [12,19,23], and PRM-VS-TF [21]. Improved decoder layers in the proposed model have delivered the highest F1-score (81.44%), better than the results from the baselines i.e. 79.82%, 78.08%, and 77.66%, respectively. Recall exhibits the highest score with 81.11%, indicating the more influential class discrimination. For example, water class in Figure 6f reveals a similar pattern as found in the label image and the comparative PRM-VS-TF (Figure 6b,e). Inconsistent channel results are found in DeepLabV3 and SwinTF (Figure 6d,e). A drawback was witnessed in the precision score when the proposed model revealed 0.96 less performance than the PRM-VS-TF (82.75% vs. 81.79%). This result was expected when the segmented result delivers a high recall score. Nevertheless, there was no influence on the remotely sensed image segmentation. On the other hand, the proposed E-PRM-VS-TF can overcome the other two comparative models. In Table 2, E-PRM-VS-TF outperforms all models when considering the average accuracy of each segmented class.

**Table 1.** Effect on the testing dataset of Thailand Landsat-8 challenge dataset.

|          | Frontend     | Model              | *Precision* | *Recall*   | *F1*       |
|----------|--------------|--------------------|-------------|------------|------------|
| Baseline | DenseNet-201 | DeepLabV3 [8]      | 79.27%      | 76.38%     | 77.66%     |
|          | ViT          | SwinTF [12,19,23]  | 78.49%      | 77.71%     | 78.08%     |
|          | ViT          | PRM-VS-TF [21]     | **82.75%**  | 78.87%     | 79.82%     |
| **Proposed** | ViT      | E-PRM-VS-TF        | 81.79%      | **81.11%** | **81.44%** |

**Table 2.** Effects on the testing dataset of the Thailand Landsat-8 challenge for each class with the proposed procedures in terms of *Average Accuracy*.

|                     | Model             | Agri       | Forest     | Misc       | Urban      | Water      |
|---------------------|-------------------|------------|------------|------------|------------|------------|
| Baseline            | DeepLabV3 [8]     | 86.78%     | 89.67%     | 66.48%     | 82.64%     | 66.90%     |
|                     | SwinTF [12,19,23] | 86.76%     | 88.94%     | 73.39%     | 91.98%     | 72.70%     |
|                     | PRM-VS-TF [21]    | 87.89%     | 88.91%     | 76.65%     | 93.31%     | 74.48%     |
| **Proposed Method** | E-PRM-VS-TF       | **89.83%** | **89.28%** | **79.68%** | **94.50%** | **75.50%** |

In Figures 4 and 5, training accuracy and the loss values of both PRM-VS-TF and our proposed E-PRM-VS-TF are displayed. The proposed E-PRM-VS-TF demonstrates smoother curves in training and validation (Figure 5) compared to the baseline (Figure 4). The enhanced decoder layers can improve the upsampling performance, expanding the feature's original resolution, resulting in precise feature detection and higher segmentation accuracy.

In Figures 6–8, the segmentation sub-scenes of the Thailand Landsat 8 corpus of the proposed and baseline models are provided. Each row in the figures represents the test-set samples. Column a illustrates the input and column b illustrates the label image. The baseline models (DeepLabV3, Swin-TF, and PRM-VS-TF) and our proposed model (E-PRM-VS-TF) are subsequently represented in columns c, d, e, and f. In Figure 6, the focus is on urban and water classes. The prediction from the proposed model indicates higher potential than its counterparts. Notice the urban class segmentation in the middle

row: E-PRM-VS-TF (f) predicted a similar result to the label image (b). The water class also produced an identical outcome, as seen in the label image; the channels are smoothly merged. Other channel predictions display some inconsistent predictions.

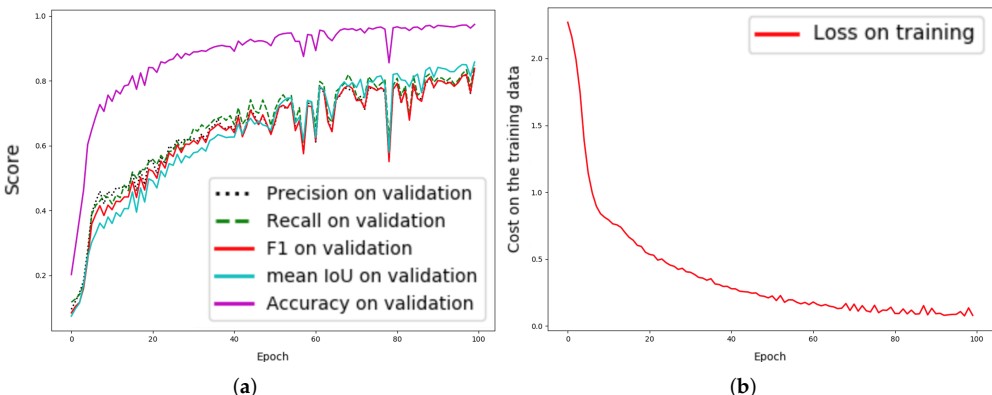

(**a**)                                                                                    (**b**)

**Figure 4.** The learning curves of the baseline, "PRM-VS-TF", on the Thailand Landsat-8 corpus; x refers to epochs and y refers to different measurements: (**a**) plot of model loss (cross-entropy) from training and validation corpora and (**b**) performance plot on the validation corpus.

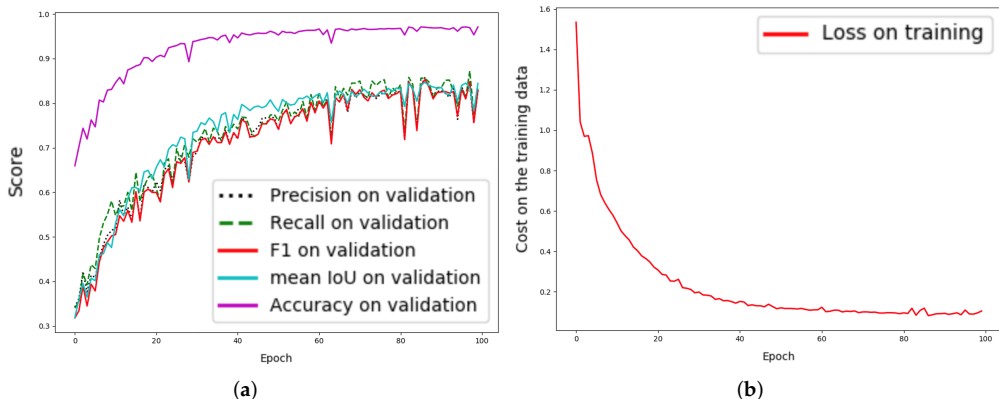

(**a**)                                                                                    (**b**)

**Figure 5.** The learning curves for the proposed approach, "E-PRM-VS-TF", on the Thailand Landsat-8 corpus; x refers to epochs and y refers to different measurements: (**a**) plot of model loss (cross-entropy) on training and validation corpora and (**b**) performance plot on the validation corpus.

As shown in Figure 7, all models performed well with the forest class, since they exhibit the same pattern, as shown in the label image. In Table 2, *Average Accuracy* results are identical to the predictions in Figure 7, revealing no significant difference. As for column f, E-PRM-VS-TF reveals higher performance along the edge in the segmented area.

The miscellaneous class (row 3 in Figure 8) was well predicted by the proposed model (column f) and outperformed all replications. Concerning the results from DeepLabV3 and Swin-TF (row 3, columns c and d), confusions of segmentation are shown. PRM-VS-TF, i.e., column e, was able to produce proper segmentation for the miscellaneous class, although the prediction for the urban class was still inadequate.

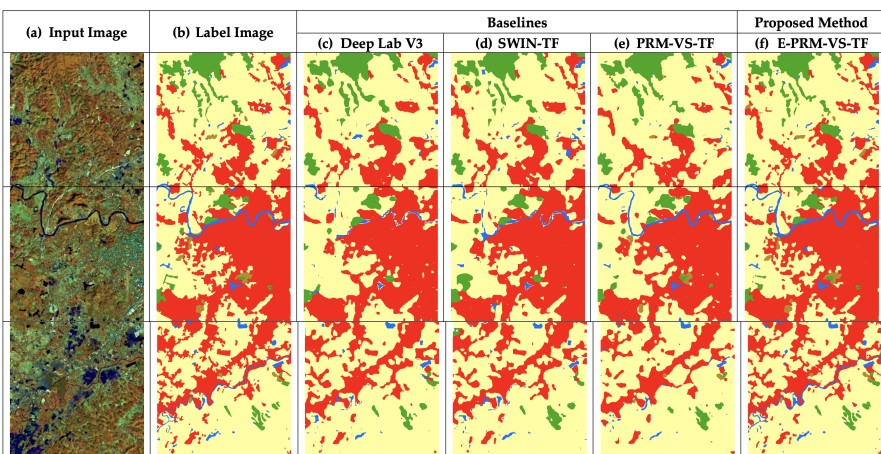

**Figure 6.** Comparison between the proposed method and baselines: focusing on the urban (red) and water (blue). The proposed method exhibits consistency in channels' consequences.

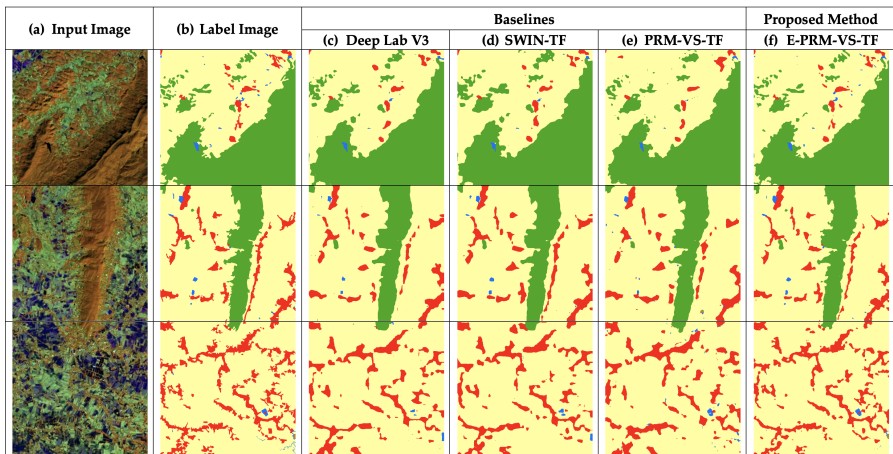

**Figure 7.** The performance of the forest (green) prediction reveals no significant difference between any models.

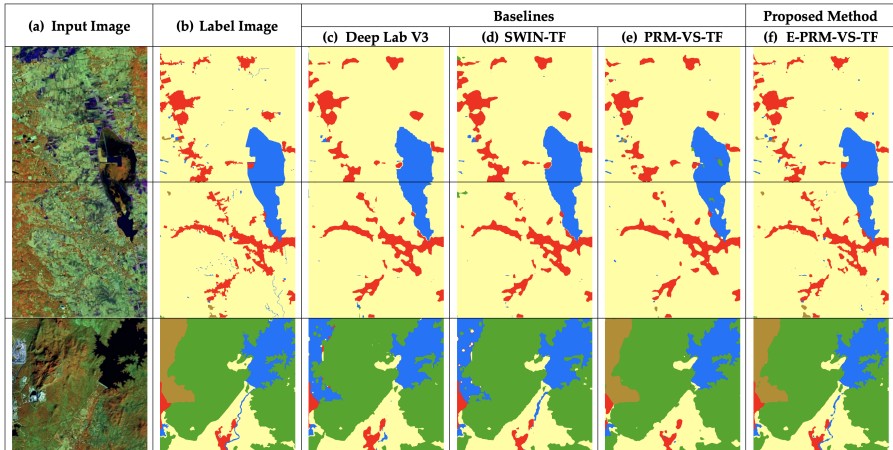

**Figure 8.** In the first two rows, the proposed method demonstrates sufficient performance for the urban predictions (red). In the third row, the miscellaneous results (brown) predicted by the proposed model (column **f**) outperform all replications.

In Figures 9 and 10, the entire Landsat-8 corpus segmentation scene representing the north-eastern and central regions predictions of E-PRM-VS-TF is shown. All classes were precisely predicted, and were much improved from previous counterparts' segmentations.

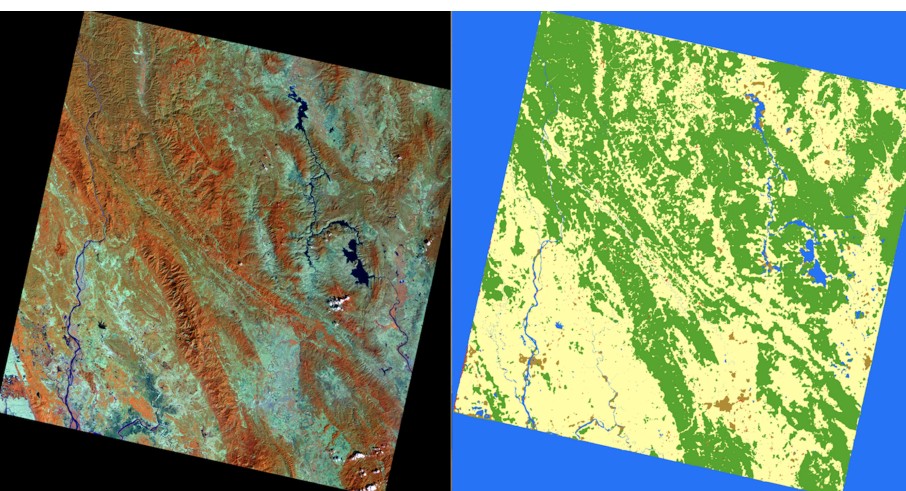

**Figure 9.** The entire scene's prediction by "E-PRM-VS-TF" displays the forest and agriculture distinctions in the northern region of Nan Province.

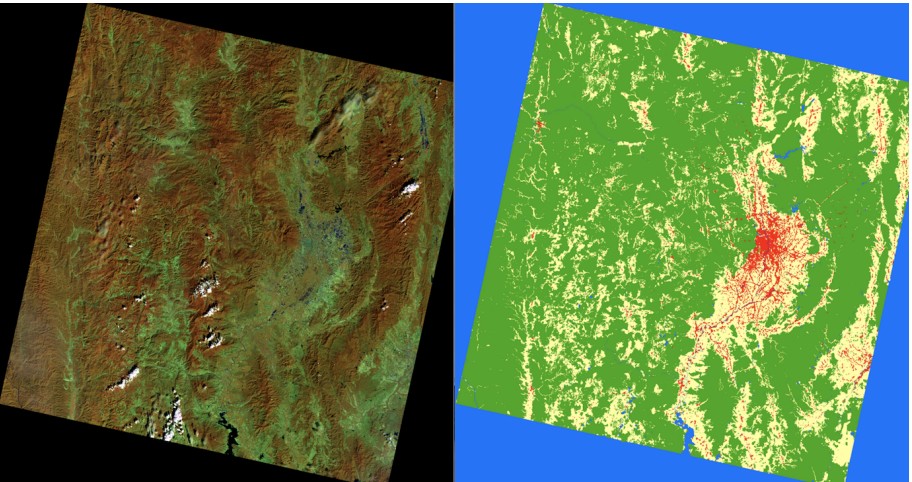

**Figure 10.** The zoomed predictions of the proposed method distinguishing urban from agriculture in the central region of Nan Province.

Another ingrained limitation of transformers in sequential tasks is the lack of recursive computation, and the deep model depth bounds the number of transformations possible on the input. Such drawbacks impact tasks that require careful tracking of a world state or modeling hierarchical structures.

## 5. Conclusions

In this study, E-PRM-VS-TF, an enhanced transformer backbone for dense prediction of semantic segmentation tasks was implemented. The E-PRM-VS-TF architecture can refine features and deliver more coverage, increasing the accuracy of agriculture, forest, miscellaneous, urban, and water classes. Hence, both the progressive shrinking pyramid transformer's decoder and the spatial-reduction attention layer were much improved, enabling it to achieve higher resolution and multi-scale feature maps. Extensive experiments on semantic segmentation benchmarks verified that our model is more robust than the well-designed CNNs under equivalent parameters. Results revealed that the "E-PRM-VS-TF" model significantly transcended all baseline F1 score. It was Thailand's Landsat-8 challenge

dataset winner, surpassing 80% in F1-scores. Moreover, E-PRM-VS-TF achieved accuracy transcending 89% in almost all classes.

**Author Contributions:** Conceptualization, T.P.; Formal analysis, T.P.; Investigation, T.P.; Methodology, T.P.; Project administration, T.P.; Resources, T.P.; Software, T.P.; Supervision, T.P.; Validation, T.P.; Visualization, T.P.; Writing–original draft, T.P.; Writing–review and editing, T.P., K.I.; Applaud, P.R. ; All authors have read and agreed to the published version of the manuscript.

**Funding:** The Office of Thailand Science Research and Innovation (TSRI) is gratefully acknowledged for their financial support under contract no. RDG6130016. This research was supported by NASA Land Cover Land Use Change Grant. It was also funded by the CGIAR Global Rice Science Partnership (GRiSP) program.

**Institutional Review Board Statement:** Not applicable.

**Informed Consent Statement:** Not applicable.

**Data Availability Statement:** Not applicable.

**Conflicts of Interest:** The authors declare no conflict of interest.

## Abbreviations

The following abbreviations are used in this manuscript:

| | |
|---|---|
| CNNs | Convolutional Neural Networks |
| DeiT | Data-efficient Image Transformers |
| DL | Deep Learning |
| E | Enhanced |
| Param | Parameters |
| PRM | Pyramid |
| PRM-VS-TF | Pyramid Vision Transformer |
| R-CNN | Region-based Convolutional Neural Networks |
| SwinTF | Swin Transformer |
| TU | Transitions Up |
| TD | Transitions Down |
| ViT | Vision Transformer |

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
