# Peer review of "Enhanced Feature Pyramid Vision Transformer for Semantic Segmentation on Thailand Landsat-8 Corpus"

_information, doi:10.3390/info13050259_

Round 1

Reviewer 1 Report

The manuscript proposes a novel network for semantic segmentation on a specific dataset. The authors did this by incorporating the attention mechanism across the network feature maps. Although the experiments support the authors' claims, I still have the following concerns for publication of this manuscript:

  1. Is it possible to highlight the novel modules within the network in Figure 2? I suppose the novel parts lie in 'multi-head attention'? Further, is it possible to show a typical network without the attention mechanism, to help readers differentiate from your scheme?
  2. Similar to Point 1, it would be great to incorporate a short description of what the attention layers learn based on the observation of Figure 3? Figure 3 in its current form does not provide useful information. You have incorporated the computation procedure of the attention in your manuscript, which is good. 

Author Response

Our revised paper with highlights and responding to reviewers' comments: https://drive.google.com/drive/folders/1L8aAMsOR8-wc3bGuf__QkvrcEh19QeRF?usp=sharing

Q1) The manuscript proposes a novel network for semantic segmentation on a specific dataset. The authors did this by incorporating the attention mechanism across the network feature maps. Although the experiments support the authors' claims, I still have the following concerns for the publication of this manuscript:

A1) We appreciate your detailed comments and suggestions. There are identified some essential points that we hope to clarify and address here and in our revision.

Q2) Is it possible to highlight the novel modules within the network in Figure 2? I suppose the novel parts lie in 'multi-head attention'? Further, is it possible to show a typical network without the attention mechanism, to help readers differentiate from your scheme?

A2) Yes, we have highlighted the novel modules within the network in Figure 2 with a yellow box through SwinTransformer layers and we also removed the attention mechanism.

Q3) Similar to Point 1, it would be great to incorporate a short description of what the attention layers learn based on the observation of Figure 3? Figure 3 in its current form does not provide useful information. You have incorporated the computation procedure of the attention in your manuscript, which is good.

A3) We have already revised by rewriting as Figure 3. In the visualization of attention, each cell reveals a cosine similarity between its embedding and other pairwise embeddings.

Reviewer 2 Report

This paper proposed an enhanced Pyramid Vision Transformer network for semantic segmentation on the Thailand Landsat-8 dataset. Experiment results show that the proposed network has outstanding performance on the Thailand Landsat-8 dataset compared to some baseline methods.

Some detailed suggestions or problems of this paper are listed below.

1. It seems that this manuscript is not very carefully prepared. In section 1 introduction, the author introduces the structure of this paper as “Section 2 addresses related studies, while Section 3 details our data collection. Following that, Section 4 delves into the specifics of our technique, and Section 5 illustrates the findings of the performance’s investigation. Finally, in section 6, conclusions are formed.” Yet it is totally different from the actual organization of this paper. And the arrangement introduced in section one is much better than the actual structure of this paper.

2. The innovation and contribution of this paper are not well summarized and analyzed. Some essential introduction and analyses are not clear enough in both the introduction and the experiment analysis part, such as: what is the shortage of the PVT or SWIN TF and what is the idea of this paper to overcome this shortage, how is the proposed method to achieve this idea as an enhancement to the PVT, some additional experiments or ablation studies to demonstrate the idea and so on.

3. Table 1 is too simple which makes it seems unnecessary. Some more details of the dataset could be added, or it can be clearly described with one sentence in the context.

4. The introduction of the proposed method is not good enough.

Too many texts for generally introducing existing methods are not suitable in the section that was supposed to introduce the method of this paper.

As the only two figures in the method section, Figure 2 and 3 are not explained or described.

The content of Figure 2 does not reflect those method introduction paragraphs well. There seems to be no relationship between them.

Some more figures to illustrate the method details are recommended for readers’ better understanding.

5. If there is no subsection 2.2.2, the subsection of 2.2.1 looks strange, and it is about training details, which should be put in the experiment section.

6. The hardware and software environment for the experiments should be introduced.

7. It seems that this paper is based on a competition for the Thailand Landsat-8 dataset, so it is the only adopted dataset.

If the network is specially designed for this dataset, then the specialty of the network or the dataset should be analyzed.

If it is a network adapted to semantic segmentation for generalized aerial and satellite images, more experiments on public datasets would make the performance more convincing.

Author Response

Our revised paper with highlights and responding to reviewers' comments: https://drive.google.com/drive/folders/1L8aAMsOR8-wc3bGuf__QkvrcEh19QeRF?usp=sharing

Q1) This paper proposed an enhanced Pyramid Vision Transformer network for semantic segmentation on the Thailand Landsat-8 dataset. Experiment results show that the proposed network has outstanding performance on the Thailand Landsat-8 dataset compared to some baseline methods. Some detailed suggestions or problems of this paper are listed below.

A1) We appreciate your detailed comments and suggestions. There are identified some essential points that we hope to clarify and address here and in our revision. Sorry for any indistinction. As mentioned in the general rebuttal, we agree with this unclear.

Q2) It seems that this manuscript is not very carefully prepared. In section 1 introduction, the author introduces the structure of this paper as “Section 2 addresses related studies, while Section 3 details our data collection. Following that, Section 4 delves into the specifics of our technique, and Section 5 illustrates the findings of the performance’s investigation. Finally, in section 6, conclusions are formed.” Yet it is totally different from the actual organization of this paper. And the arrangement introduced in section one is much better than the actual structure of this paper.

A2) We have strictly followed all suggestions. If there are any further improvements needed, please let us know. Moreover, we have fully revised the structure of the paper with a native English speaker to improve the paper's writing quality significantly. The remainder of this article is structured as follows. Section 1 presents an introduction. Section 2 discusses the data collection.  The proposed method is detailed in Section 3, and Section 4 presents our experimental results, including the hardware and software environment, limitations, and outlook. Finally, our conclusions are drawn in Section 5.

Q3) The innovation and contribution of this paper are not well summarized and analyzed. Some essential introduction and analyses are not clear enough in both the introduction and the experiment analysis part, such as: what is the shortage of the PVT or SWIN TF and what is the idea of this paper to overcome this shortage, how is the proposed method to achieve this idea as an enhancement to the PVT, some additional experiments or ablation studies to demonstrate the idea and so on.

A3) We have revised this part by adding into subsection “Pyramid Vision Transformer Based Semantic Segmentation”: The traditional SwinTF model \citep{liu2021swin} focuses on the relationship between a token (image patches); the other tokens are calculated. PVT focuses on the quadratic complexity concerning the number of image patches; finding it unsuitable for many image problems requiring an immense set of tokens for the softmax layer.

Q4) Table 1 is too simple which makes it seems unnecessary. Some more details of the dataset could be added, or it can be clearly described with one sentence in the context.

A4) Thank you for the suggestion. We have removed Table 1 and described it in one sentence.

Q5) The introduction of the proposed method is not good enough. Too many texts for generally introducing existing methods are not suitable in the section that was supposed to introduce the method of this paper. As the only two figures in the method section, Figures 2 and 3 are not explained or described. Some more figures to illustrate the method details are recommended for readers’ better understanding.

A5) Thank you for your asking. We have revised the manuscript and modified it accordingly. We have corrected the description in figure 3. -- Figure 3. In the visualization of attention, each cell reveals a cosine similarity between its embedding and other pairwise embeddings.

Q6) If there is no subsection 2.2.2, the subsection of 2.2.1 looks strange, and it is about training details, which should be put in the experiment section.

A6) We have removed both subsections 2.2.1 and subsection 2.2.2.

Q7) The hardware and software environment for experiments should be introduced.

A7) We have revised this part by adding into subsection “Experimental Results”: Regarding the deep learning (DL) environmental setup, the “PyTorch v0.12” was created as an end-to-end open-source platform. All experiments were carried out via servers with Intel® Xeon® Scalable 4210R (10 core, 2.4 GHz, 13.75 MB, 100W), 256 GB of memory, and the NVIDIA GeForce RTX™ 2080 Ti Integrated with 11GB GDDR6 352-bit memory interface × 2 cards.

Q8) It seems that this paper is based on a competition for the Thailand Landsat-8 dataset, so it is the only adopted dataset. If the network is specially designed for this dataset, then the specialty of the network or the dataset should be analyzed. If it is a network adapted to semantic segmentation for generalized aerial and satellite images, more experiments on public datasets would make the performance more convincing.

A8) In this study, several experiments have been conducted and the results show the same reports. But the article shows the experiment in Thailand because we have complete training and testing dataset. This research used data from the Department of Land Development, which is the standard information in the country. We also tested the use of open data in our experiments as shown.

Reviewer 3 Report

In the Abstract, the authors said that “the model may outperform current state-of-the-art results in regular benchmarks for assessing farm and forest aspects.” Why for the urban class, water class, miscellaneous class, the proposed model may not outperform?

Does the proposed method have some shortcomings?

In the experiments, the results of the proposed method should also be compared with the original transformer method.

At the end of Introduction part, it is better to add a paragraph to highlight the contribution of the article.

Figure 3, what’s the meaning of the sub-figure and what’s the meaning of the x-y axis?

Figures 6,7,8, all the figure titles are the same.

Figures 9-10, what is the meaning of the sub-figures?

Too many English abbreviations do not give full English names on the first using. PVT, DL, R-CNN, DeiT, SwinTF etc.

Author Response

Our revised paper with highlights and responding to reviewers' comments: https://drive.google.com/drive/folders/1L8aAMsOR8-wc3bGuf__QkvrcEh19QeRF?usp=sharing 

Q1) In the Abstract, the authors said that “the model may outperform current state-of-the-art results in regular benchmarks for assessing farm and forest aspects.” Why for the urban class, water class, and miscellaneous class, the proposed model may not outperform?

A1) We sincerely appreciate your additional feedback. Surprise in the results, we agree that the fact that the urban class, water class, and miscellaneous class, the proposed model may not outperform the baseline, however; we rely on the overall performance as the quality of the model.

Q2) Does the proposed method have some shortcomings?

A2) Yes, we have some shortcomings in the proposed method and we have added this into the last paragraph in Subsection Experimental Results. “Another ingrained limitation of Transformers on sequential tasks is the lack of recursive computation, and the deep model depth bounds the number of transformations possible on the input. Such drawbacks impact tasks that require careful tracking of a world state or modeling hierarchical structures.”

Q3) In the experiments, the results of the proposed method should also be compared with the original transformer method.

A3) Thanks to the reviewer very much for the comment. We have reported the comparison in Tables 1 and 2, row 2.

Q4) At the end of the introduction part, it is better to add a paragraph to highlight the contribution of the article.

A4) Thank you for your suggestion. We have placed it on page 2, line 58.

Q5) Figure 3, what’s the meaning of the sub-figure, and what’s the meaning of the x-y axis?

A5) Thank you for your comment. Figure 3. In the visualization of attention, each cell reveals a cosine similarity between its embedding and other pairwise embeddings.

Q6) In Figure 6,7,8, all the figure titles are the same.

A6) Thank you for asking, we have corrected the descriptions of the figures: Figure 6. Comparison between the proposed method and the baselines focusing on the urban (red) and water (blue); the proposed method exhibits the consistency of channels' consequences Figure 7. The performance of the forest (green) prediction reveals no significant difference between all models Figure 8. In the first two rows, the proposed method demonstrates a sufficient performance of the urban's prediction (red); In the third row, the miscellaneous (brown) predicted by the proposed model (column f) outperforms all replications.

Q7) Figures 9-10, what is the meaning of the sub-figures?

A7) Thank you for asking, we have corrected the descriptions. Figure 9. The entire scene's prediction of “E-PRM-VS-TF” display the forest and agriculture distinction in the northern region of Nan Province Figure 10. The concentrated prediction of the proposed method to identify the urban from agriculture that is surrounded by the forest in the central region of Nan Province.

Q8) Too many English abbreviations do not give full English names on the first use. PVT, DL, R-CNN, DeiT, SwinTF, etc.

A8) We have revised the English names in Subsection Abbreviations. Thank you for your suggestion.

Round 2

Reviewer 2 Report

The questions and problems are answered and revised. I have no more questions.

Reviewer 3 Report

no